# A correlation analysis between nutritional status and physical function in the elderly with pulmonary tuberculosis

Qiaolin Yu[1☉], Rong Yao[2☉], Limei Lei[2], Xiaoli Shao[1], Leilei Huang[2], Fanghui Xie[2], Yan Zhou[2], Ting Zhang[1], Yuanyuan Li[1], Xiang Long[3], Miao Zhang[3], Xiaoyi Yang[2]*, Yinping Hu[1]*

**1** The Frist Department of Tuberculosis, the Public Health Clinical Center of ChengDu, Jinjiang District, Chengdu, Sichuan, China, **2** The Second Department of Tuberculosis, the Public Health Clinical Center of ChengDu, Jinjiang District, Chengdu, Sichuan, China, **3** The Department of Nursing, the Public Health Clinical Center of ChengDu, Jinjiang District, Chengdu, Sichuan, China

☉ These authors have contributed equally to this work and share first authorship.
* 1365390609@qq.com (XY); 465155357@qq.com (YH)

## Abstract

### Background

Malnutrition is a significant risk factor contributing to the progression of the elderly with pulmonary tuberculosis (TB). This study aimed to evaluate the nutritional status of the elderly with pulmonary TB using the Mini Nutritional Assessment (MNA) and explore the relationship between their nutritional status and physical function.

### Methods

This was a cross-sectional survey study. Data collection was from July 2023 to March 2024. 532 the elderly with pulmonary TB who were admitted to a tertiary infectious disease hospital in Chengdu were included in this analysis. The nutritional status of the patients was evaluated using MNA, and they were divided into well-nourished group (≥24 points) and abnormal-nourished group (<24 points). This study assessed physical function using the Berg Balance Scale (BBS), the Timed Up and Go test (TUG), and the Five-Times-Sit-to-Stand Test (FTSST). Clinical data and physical function of the two groups were compared, and the correlation between nutritional score and physical function was analyzed.

### Results

There were 109 cases (20.5%) in well-nourished group and 423 cases (79.5%) in abnormal-nourished group. Compared with well-nourished group, the abnormal group showed a decrease in the BBS scores [(52.55 ± 7.10) vs (43.20 ± 16.29), $p < 0.05$], and an increase in the TUG [9.00 (7.00, 10.00) vs 9.00 (7.40, 12.00), $p < 0.05$] and the FTSST [12.00 (9.00, 14.75) vs 15.00 (10.00, 20.10), $p < 0.05$]. Correlation

**Data availability statement:** All relevant data are within the paper and its Supporting Information files.

**Funding:** The work was funded by Health Commission of Sichuan Province Medical Science and Technology Program (24QNMP049) and the Chengdu Municipal Health Commission Subjects (2023185), with YY L and QL Y as the recipients of the funding, respectively. The funders had no role in study design, data collection and analysis, decision to publish, or preparation of the manuscript.

**Competing interests:** The authors declare that they have no competing interests.

analysis showed that the nutritional score of patients was positively correlated with the BBS scores ($r = 0.474$, $p < 0.001$), and negatively correlated with the TUG ($r = -0.200$, $p < 0.001$) and the FTSST ($r = -0.501$, $p < 0.001$).

## Conclusions

Malnutrition is common in the elderly with pulmonary TB. Nutritional status in these patients is associated with the BBS scores, the TUG, and the FTSST.

---

## Background

Tuberculosis (TB) remains one of the major infectious diseases threatening human health [1]. According to the Global Tuberculosis Report 2024 [2], there were approximately 10,800,000 new TB cases worldwide in 2023, with an incidence rate of 134/100,000 population. In the same year, China reported approximately 741,000 new TB cases, with an incidence rate of 52/100,000, accounting for 6.8% of the global total, ranking third in the world, after India (26%) and Indonesia (10%). With the intensification of population aging, the elderly has become a high-risk group for TB due to their physiological characteristics [3]. Studies have shown that the reported incidence of pulmonary TB among the elderly in China was 2.4–2.9 times higher than that of the non-elderly between 2011 and 2020 [4]. Moreover, in 2021, the proportion of the elderly with pulmonary TB aged 65 and above reached as high as 28.9% [5]. Therefore, the role of the elderly in pulmonary TB prevention and control has garnered significant attention.

Due to the decline in physiological function, reduced immune capacity, and the high metabolic demands of the disease itself, malnutrition is a prominent issue among elderly TB patients [6]. Studies have shown that the overall prevalence of malnutrition among TB patients ranges from 38.3% to 75.0% [7,8], while the risk of malnutrition among elderly patients with TB is as high as 62.7% [9]. Substantial evidence indicates that malnutrition is one of the critical risk factors contributing to the onset and progression of TB [10,11]. Furthermore, malnourished elderly patients with TB are more prone to issues such as weight loss, organ function decline, frailty, and even cachexia [12]. These conditions not only significantly impair patients' quality of life but also increase the risk of adverse events such as falls, thereby severely affecting patient prognosis [13,14]. Studies have shown that physical function is an important criterion for predicting the health status of the elderly patients and a significant factor in assessing nutritional risk [15,16]. In light of this, nutritional interventions and improvements in physical function for the elderly with pulmonary TB should be integral components of TB prevention and control efforts, aiming to enhance patients' overall health and quality of life.

However, current research on the nutritional status of TB patients primarily focuses on different age groups, with relatively limited systematic studies on the nutritional status and predictive indicators of the elderly with pulmonary TB. Therefore, this study employed the Mini Nutritional Assessment (MNA) to evaluate the nutritional

status of elderly patients with pulmonary TB and further explored its relationship with physical function. The findings aimed to provide a reference for improving the health status of the elderly with TB, optimizing nutritional intervention strategies, and enhancing disease outcomes.

## Materials and methods

### Study design

This was a cross-sectional survey study (see S1 Table). Data collection was from July 2023 to March 2024.

### Participants and procedures

A convenience sampling method was used to select a total of 532 the elderly with pulmonary TB from a tertiary infectious disease hospital in Sichuan Province. Inclusion criteria: (1) Age ≥ 60 years old. (2) Comply with the Diagnosis of Tuberculosis (WS 288–2017) [17]. (3) Clear consciousness, normal communication and written consent (see S2 Table). Exclusion criteria: patients with musculoskeletal disorders, malignant tumours, chronic haemorrhage, cardiac, hepatic, renal and other vital organ insufficiency. The sample size was designed based on Kendall's Advanced Theory of Statistics [18], the minimum sample size should be at least five times the number of variables. There were 21 variables in this study with a sample size of at least 210 patients. A total of 550 questionnaires were distributed, resulting in 532 valid responses after removing invalid submissions (due to time constraints or lack of interest), with an effective recovery rate of 96.72%.

### Research tools

**General information questionnaire.** The questionnaire was designed by the researcher based on the research objectives and consisted of 15 entries. Demographic data included 11 items: gender, ethnicity, age, education, marital status, occupation, residence, monthly personal income, number of people in the household, smoking status, and alcohol consumption status. Disease information included four items: number of co-morbidities, history of TB, treatment status, and body mass index (BMI).

**Mini nutritional assessment (MNA).** The assessment was an internationally recommended nutritional screening tool for elderly patients [19–21]. It consisted of anthropometric indicators (4 entries), overall assessment (6 entries), dietary assessment (6 entries) and subjective rating (2 entries). There were 18 entries with a total score of 30.The judgement criteria was well nourished: ≥ 24 points, risked malnutrition: 17 to <24 points, malnourished: < 17 points. The scale demonstrates effective screening performance for hospitalized elderly patients with chronic diseases [22]. Based on clinical experience and relevant literature [23,24], all TB patients with nutritional risk should be assessed for their nutritional status. Therefore, in our study, patients with MNA scores <24 points (i.e., risked malnutrition: 17–24 and malnourished: < 17 points), were categorised as abnormal-nourished group [25]. (see S2 Table).

**Berg Balance Scale (BBS).** The scale consisted of 14 balance-related entries as a means of assessing functional body balance in older adults [26]. A 4-point Likert scale (0 indicating the lowest level of functions and 4 indicating the highest) was used, with a total score of 56. 0–20 points (poor balance ability and can only sit in a wheelchair), 21–40 points (balance ability and can assist with walking), 41–56 points (good balance ability and can walk independently) [27]. The Cronbach's alpha of the scale was 0.864. (see S2 Table)

**Timed Up and Go test (TUG).** The test is recommended by the American Geriatrics Society as an indicator for assessing the patient's balance and walking function, especially for the elderly [28]. Test method: During the test, the patients wore their usual shoes, sat on a chair with armrests and a backrest, leaned their body on the back of the chair, and placed their hands on the armrests (seat height about 45 cm, armrest height about 20 cm). Stick or place a visible marker on the floor 3 metres away from the chair. When the command "start" was heard, the patient stood up firmly, followed the usual walking gait, walked towards the marker, turned around, returned to the chair, sat down, and leaned

against the back of the chair. The researcher recorded the time it took the patient to complete the entire walking process, with the shorter time indicating the patient's ability to walk and balance better. The criteria were as follows: < 10s was well, suggesting free movement; ≥ 10s was adverse, suggesting impaired movement [29].

**Five-Times-Sit-to-Stand Test (FTSST).** This test, which recorded the time it took the subject to repeatedly stand up and sit down five times from a chair, was commonly used to assess lower limb strength and balance in the elderly [30]. Test method: The patients crossed their arms in front of their chests, looked straight ahead, and stood up and sat down 5 times as fast as they could from a 46 cm high chair. All patients repeated the test 3 times with a 1 min break in between, and the average of the 3 times was taken as the final test result. Finally, the average test time for all patients was 10s. Thus, Time ≥ 10s was considered risk of falling; < 10s was considered good balance and no risk [31].

## Data collection

The researcher used a standardized guideline to explain the study to the patients after obtaining their consent. Simultaneously, an electronic questionnaire and a QR code were generated for independent participation by scanning the code. In instances where patients were unable to complete the questionnaire, the researcher assisted them by dictating responses and filling it on their behalf, ensuring data completeness. Raw data was collected using the Questionnaire Star platform. Duplicate questionnaires with identical addresses and content were excluded post-survey. Disease-related information was extracted from medical records. Prior to the study, researchers underwent theoretical, operational, and post-training assessments.

## Statistical analysis

SPSS 21.0 was used to analyse the data (see S1 File). All continuous variables were tested for normality using the Shapiro-Wilk test ($\alpha = 0.05$). Variables that conformed to normal distribution (Berg Balance Scale) were described using mean ± standard deviation, and t-tests were used for comparisons between groups. Variables that did not fit the normal distribution (Timed Up and Go test and Five-Times-Sit-to-Stand Test) were described by median [IQR], and comparisons between groups were made using the two related samples rank-sum test. Count data were described by n (%), and comparisons between groups were made using the $\chi^2$ test. Correlations were analyzed using spearman's correlation analysis, and a difference of $P < 0.05$ was considered statistically significant.

## Ethical issues

This study was approved by the Medical Ethics Committee of Public Health Clinical Centre of Chengdu (NO YJ-K2023-20–01). Written informed consent was obtained from participants with signing capacity. Those without signing capacity, family consent and informed consent were also obtained after explaining the study process to them.

## Results

### Nutritional status of the elderly with pulmonary TB

The nutritional status score of 532 patients ranged from 8 to 30. The number of cases in each group and nutritional status scores were shown in Table 1.

Table 1. Nutritional status of the elderly with pulmonary TB.

| Groups | n(%) | Score |
|---|---|---|
| well-nourished | 109(20.5) | 25.61 ± 1.45 |
| abnormal-nourished | 423(79.5) | 18.08 ± 4.01 |
| total nutritional score | / | 19.62 ± 4.74 |

### The comparison of general information between the two groups

There were 109 cases of well-nourished patients, with mean age $67.94 \pm 5.80$. There were 423 cases of abnormal-nourished patients, with an average age of $70.34 \pm 6.96$. The differences between the two groups in terms of gender, age, history of alcohol consumption, BMI, Berg Balance Scale, Timed Up and Go test and Five-Times-Sit-to-Stand Test were statistically significant, as shown in Table 2.

### The comparison of physical function between the two groups

The scores of Berg Balance Scale, Timed Up and Go test, and Five-Times-Sit-to-Stand Test were compared between the 2 groups, and the differences were statistically significant ($P < 0.05$), as shown in Table 3.

### Analysis of the correlation between nutritional status and physical function

Nutritional status was positively correlated with Berg Balance Scale in elderly tuberculosis patients and negatively correlated with the Timed Up and Go test and the Five-Times-Sit-to-Stand Test, as shown in Table 4.

## Discussion

The nutritional status influences the occurrence and progression of TB in the elderly. Our study investigated the clinical indicators and physical function differences between elderly TB patients with good nutritional status and those with abnormal nutritional status, aiming to provide a theoretical foundation for the clinical prevention and treatment of such patients.

### Risk of malnutrition prevalent in the elderly with pulmonary TB

The incidence of malnutrition in hospitalized patients has been reported to be 15% to 60% [32]. However, the elderly with pulmonary TB could be at a higher nutritional risk due to multiple admissions to hospitals for long-term anti-tuberculosis treatment. In our study, the incidence of malnutrition risk among elderly patients with pulmonary TB was 79.5%, which was similar to the findings of Magassouba et al [33] in 218 drug-resistant TB patients in Guinea. However, it was much higher than the study of NISHIOKA et al [34], where the risk of malnutrition in 420 elderly stroke patients was 29.76%. This may be due to the long-term oral anti-TB drugs in the elderly with TB, heavy gastrointestinal and other adverse reactions, weak metabolism, insufficient nutrient absorption, as well as intake, coupled with the increased disease burden on the body, which can easily lead to malnutrition in patients. Therefore, health care personnel should pay attention to the nutritional status of the elderly with TB during diagnosis and treatment, formulate long-term and effective nutritional support programs, improve their nutritional status, enhance their immunity and resistance, alleviate their clinical symptoms, shorten their hospitalization time, and thus effectively improve the efficiency of clinical treatment.

### Potential factors influencing the nutritional status of elderly tuberculosis patients

We compared the clinical characteristics of elderly tuberculosis patients in the well-nourished group and the abnormal-nourished group, and found significant differences between the two groups in terms of gender, age, alcohol consumption, BMI, and three physical function tests. These differences are discussed in detail as follows:

(1) Gender: In this study, women had poorer nutritional status than men, which was consistent with the results of a large sample survey [35]. This may be due to the fact that metabolism in men's bodies is faster than in women's, and they have a greater ability to absorb nutrients, thus being better nourished.

(2) Age: Our study found that among elderly TB patients (aged >75), the proportion of cases with poor nutritional status was significantly higher than those with good nutritional status. Several factors may contribute to this finding. First, the physical function and metabolism of elderly individuals gradually decline, leading to physiological deterioration

**Table 2. Comparison of general information between the two groups.**

| Characteristics | | Groups | | χ²/t | p |
|---|---|---|---|---|---|
| | | Well-nourished(n=109) | Abnormal-nourished (n=423) | | |
| Gender | male | 86(78.9%) | 293(69.3%) | 3.924 | 0.048* |
| | female | 23(21.1%) | 130(30.7%) | | |
| Ethnic | chinese | 95(87.2%) | 356(84.2%) | 0.602 | 0.438 |
| | other | 14(12.8%) | 67(15.8%) | | |
| Age (years) | 60~75 | 97(89.0%) | 327(77.3%) | 7.315 | 0.007* |
| | >75 | 12(11.0%) | 96(22.7%) | | |
| Education level | primary school | 64(58.7%) | 299(70.7%) | 6.108 | 0.106 |
| | middle school | 31(28.4%) | 80(18.9%) | | |
| | high school | 9(8.3%) | 27(6.4%) | | |
| | college or above | 5(4.6%) | 17(4.0%) | | |
| Marital status | unmarried | 1(0.9%) | 10(2.4%) | 2.366 | 0.306 |
| | married | 93(85.3%) | 335(79.2%) | | |
| | divorced/widowed | 15(13.8%) | 78(18.4%) | | |
| Employment status | employed | 5(4.6%) | 18(4.3%) | 2.107 | 0.349 |
| | retired | 42(38.5%) | 133(31.4%) | | |
| | unemployed | 62(56.9%) | 272(64.3%) | | |
| Residence | alone | 14(12.8%) | 38(9.0%) | 1.465 | 0.226 |
| | together | 95(87.2%) | 385(91.0%) | | |
| Monthly income (RMB, Yuan) | <4000 | 84(77.1%) | 356(84.2%) | 3.672 | 0.159 |
| | 4000~8000 | 23(21.1%) | 58(13.7%) | | |
| | >8000 | 2(1.8%) | 9(2.1%) | | |
| Number of family members | 1~3 | 38(34.9%) | 138(32.6%) | 2.523 | 0.300 |
| | 4~6 | 59(54.1%) | 212(50.1%) | | |
| | ≥6 | 12(11.0%) | 73(17.3%) | | |
| Drinking status | never | 44(40.4%) | 215(50.8%) | 10.906 | 0.004* |
| | quitting | 36(33.0%) | 150(35.5%) | | |
| | yes | 29(26.65) | 58(13.7%) | | |
| Smoking status | never | 42(38.5%) | 202(47.8%) | 3.592 | 0.166 |
| | quitting | 45(41.3%) | 159(37.6%) | | |
| | yes | 22(20.2%) | 62(14.7%) | | |
| Number of co-morbidities | 0 | 41(37.6%) | 136(32.2%) | 1.203 | 0.548 |
| | 1 | 42(38.5%) | 181(42.8%) | | |
| | ≥2 | 26(23.9%) | 106(25.1%) | | |
| History of tuberculosis | <1 | 79(72.5%) | 315(74.5%) | 0.179 | 0.672 |
| | ≥1 | 30(27.5%) | 108(25.5%) | | |
| Treatment status | primary | 81(74.3%) | 299(70.7%) | 0.558 | 0.455 |
| | retreatment | 28(25.7%) | 124(29.3%) | | |
| Body mass index (BMI) | <18.5 | 24(22.0%) | 131(31.0%) | 10.657 | 0.014* |
| | 18.5≤BMI<24.0 | 58(53.2%) | 230(54.4%) | | |
| | 24.0≤BMI<28.0 | 20(18.3%) | 54(12.8%) | | |
| | ≥28.0 | 7(6.4%) | 8(1.9%) | | |
| Berg Balance Scale | poor balance | 1(0.9%) | 51(12.1%) | 21.402 | 0.000* |
| | balance | 8(7.3%) | 88(20.8%) | | |
| | good balance | 100(91.7%) | 284(67.1%) | | |

*(Continued)*

**Table 2.** (Continued)

| Characteristics | | Groups | | χ²/t | p |
|---|---|---|---|---|---|
| | | Well-nourished(n = 109) | Abnormal-nourished (n = 423) | | |
| Timed Up and Go test | Adverse (≥10s) | 98(89.9%) | 294(69.5%) | 18.609 | 0.000* |
| | Well (<10s) | 11(10.1%) | 129(30.5%) | | |
| Five-Times-Sit-to-Stand Test | <10 (s) | 29(26.6%) | 50(11.8%) | 14.983 | 0.000* |
| | ≥10 (s) | 80(73.4%) | 373(88.2%) | | |

Note: *P* values represented whether the comparison between the two groups is significant. Count variables were tested by chi-square test. Continuous variables were tested with t-test and the two related samples rank-sum test. *P < 0.05 was considered statistically significant.

**Table 3.** Comparison of physical function between the two groups.

| Item | Well-nourished (n = 109) | Abnormal-nourished (n = 423) | (t/Z) | p |
|---|---|---|---|---|
| Berg Balance Scale | 52.55±7.10 | 43.20±16.29 | t = 5.854 | 0.000 * |
| Timed Up and Go test | 9.00(7.00, 10.00) | 9.00(7.40, 12.00) | Z=−2.710 | 0.007 * |
| Five-Times-Sit-to-Stand Test | 12.00(9.00, 14.75) | 15.00(10.00, 20.10) | Z=−5.590 | 0.000* |

Note: *P* values represented whether the comparison between the two groups is significant. *P* values were derived from the t-test (t) for normally distributed variables and the two related samples rank-sum test (Z) for non-normally distributed variables. *P < 0.05 was considered statistically significant.

**Table 4.** Analysis of the correlation between nutritional status and physical function.

| Item | r | P | 95%CI |
|---|---|---|---|
| Berg Balance Scale | 0.474 | 0.000** | 0.406~0.536 |
| Timed Up and Go test | −0.200 | 0.000** | −0.282~−0.113 |
| Five-Times-Sit-to-Stand Test | −0.501 | 0.000** | −0.564~−0.431 |

Note: *P* values represented Spearman's test. *P < 0.05 was considered statistically significant.

that may result in insufficient nutrient intake [36]. Second, tuberculosis patients are prone to high metabolism and energy expenditure, which increases energy consumption and can lead to malnutrition [6]. Furthermore, nutritional risk increases with age, highlighting the need for greater attention to the nutritional risks and malnutrition issues in elderly patients [37]. Lastly, a sample study in China indicated that factors such as age, disease duration, and the presence of chronic diseases were significantly positively correlated with nutritional risk. The older the individual, the higher the likelihood of malnutrition, and this is closely related to the severity of the disease [38].

(3) Alcohol consumption: In traditional beliefs, alcohol consumption, especially excessive drinking, is known to suppress food intake and nutrient absorption. In TB patients, excessive alcohol consumption may lead to inadequate energy intake and nutrient deficiencies, particularly in elderly patients. Research has shown that alcohol consumption is associated with conditions such as low body weight and low serum albumin, both of which are signs of malnutrition [39]. These factors can exacerbate the disease burden in TB patients. However, our study showed that the proportion of alcohol-consuming patients in the abnormal-nourished group (13.7%) was lower than that in the well-nourished group (26.65%). We hypothesize that this phenomenon may reflect the association between alcohol consumption and factors such as better socio-economic conditions, improved mental health, or positive social interactions. For instance, in some social environments, alcohol consumption is often linked to higher income, better healthcare conditions, and

stronger social support, which may help elderly individuals maintain better nutritional status [40]. Additionally, alcohol may promote social interactions, alleviate anxiety, and indirectly improve appetite and overall health, thereby helping maintain a better nutritional status [41]. Furthermore, alcohol consumers may pay more attention to the diversity and nutritional balance of their diet, which could contribute to a superior overall nutritional status compared to non-drinkers [42]. However, this result needs to be further validated through additional research to better understand the relationship between alcohol consumption, nutritional status, and tuberculosis treatment.

(4) BMI: According to the "Guidelines for the Appropriate Range of Body Mass Index and Weight Management in the Elderly in China (T/CNSS 021-2023) [43]," the normal range for BMI is 18.5–24.0. A low BMI (<18.5) is typically associated with malnutrition, protein deficiency, muscle atrophy, and other issues, and such patients often face problems like impaired immune function, physical weakness, and poor response to disease treatment. Previous studies have shown that low BMI is a risk factor for TB, as it can lead to weakened immunity, increased mortality, and affect the effectiveness of anti-TB treatment [44–46]. Our study showed that the proportion of low BMI patients in the abnormal-nourished group (31.0%) was significantly higher than in the well-nourished group (22.0%), which is consistent with previous findings. A high BMI (>25), associated with obesity, has an unclear relationship with TB risk. Our study found that the proportion of high BMI patients in the well-nourished group (24.7%) was significantly higher than in the abnormal-nourished group (14.7%). There is also evidence suggesting that a very high BMI may affect the treatment efficacy of TB [6], but the specific mechanisms involved still require further investigation.

(5) The differences in the three physical function tests will be discussed in detail in the following section.

Overall, clinical practice should actively identify risk factors for malnutrition through early screening and assessment, provide targeted preventive measures, reduce risk factors for malnutrition, and maximize good nutritional function to improve the nutritional status of elderly tuberculosis patients.

## The correlation between nutritional status and physical function in elderly

Physical functional status is an important indicator of health status in the elderly and a significant predictor of malnutrition and nutritional risk. BBS is used to assess physical balance ability; TUG test is used to evaluate the ability of elderly individuals to transition from sitting to standing and walking, typically measuring dynamic balance; and the FTSTS is used to assess lower limb muscle strength and functional independence. The indicators and correlation analysis showed that the elderly in pulmonary TB with poorer nutritional status had a significant decrease in the BBS scores ($t = 5.854$, $p < 0.05$), and a significant increase in the TUG ($Z = -2.710$, $p = 0.007$) and the FTSST ($Z = -5.590$, $p < 0.05$), i.e., poorer physical functions in patients with abnormal nutrition compared to well-nourished patients. Previous studies have shown that malnutrition leads to muscle strength loss, decreased bone density, and accelerated joint function deterioration, which negatively impact balance and physical stability. This, in turn, results in longer completion times for the TUG test and poorer performance in the FTSTS test [47–49]. CHRISTA et al. also found that malnutrition was a risk factor for physical dysfunction, with people with abnormal nutrition being four times more likely to develop physical dysfunction than those who are well-nourished [50]. A randomized controlled study in the United States [51] demonstrated a strong association between malnutrition and muscle strength and functional status, indicating that increasing nutrients not only improves nutritional status but also enhances balance and physical function. Meanwhile, the WHO considers nutritional support a key factor in tuberculosis treatment [52]. What's more, exercise history also has an indirect positive effect on nutritional status. Regular physical activity not only stimulates appetite and boosts metabolism but also enhances muscle mass, which improves nutrient absorption and utilization efficiency. As a result, this leads to better performance in the aforementioned physical function tests in patients [47,49]. Therefore, healthcare professionals should conduct early screening and assessment of malnutrition, implement individualized nutritional interventions, and provide targeted preventive measures, such as nutritional supplementation and exercise guidance, to reduce risk factors for malnutrition and prevent physical function impairments.

 

## Limitations and prospects

Howere, this study has certain limitations. The survey design is single-centred, which can only address tuberculosis patients in a specific region. The specificity of the geographical population may influence the generalizability and application of this study, a factor that could be validated through multi-centre, large-sample, and future studies. Secondly, we selected inpatients, which may not fully represent the nutritional status of typical the elderly with pulmonary TB, and the sample size could be expanded further. Additionally, due to constraints in data collection, we did not systematically collect detailed information on variables such as TB drug resistance (sensitive vs. resistant), disease localization (pulmonary vs. extrapulmonary), nutritional intake, co-infections/ comorbidities. Future studies could incorporate such data more comprehensively to better assess their impact on nutritional and clinical outcomes.

## Conclusions

Our study shows that malnutrition is prevalent among in the elderly with pulmonary TB and may be associated with poorer physical function. Compared with well-nourished patients, the abnormal patients tended to have lower BBS scores and higher TUG and FTSST, though further research is needed to confirm causality. These findings highlight the potential importance of nutritional screening and assessment in this population. Early identification of malnutrition and tailored interventions may help improve physical function and quality of life, but additional studies are warranted.

## Supporting information

**S1 Table.  STROBE statement-checklist.**
(DOCX)

**S2 Table.  Patient informed consent form and research questionnaires.**
(DOCX)

**S1 File.  The analyzed data of the 532 cases of this study.**
(XLSX)

## Acknowledgments

The authors thank all the patients who participated in this study.

## Author contributions

**Conceptualization:** Qiaolin Yu, Rong Yao, Miao Zhang, Xiaoyi Yang, Yinping Hu.

**Data curation:** Qiaolin Yu, Rong Yao, Limei Lei, Xiaoli Shao, Leilei Huang, Fanghui Xie, Yan Zhou, Ting Zhang, Yuanyuan Li, Xiang Long.

**Investigation:** Qiaolin Yu, Rong Yao, Limei Lei, Xiaoli Shao, Leilei Huang, Fanghui Xie, Yan Zhou, Ting Zhang, Yuanyuan Li, Xiang Long, Miao Zhang.

**Methodology:** Xiang Long, Miao Zhang.

**Project administration:** Qiaolin Yu.

**Supervision:** Rong Yao, Limei Lei, Xiaoli Shao, Leilei Huang, Fanghui Xie, Yan Zhou, Ting Zhang, Yuanyuan Li, Xiang Long, Miao Zhang, Xiaoyi Yang, Yinping Hu.

**Writing – original draft:** Qiaolin Yu, Rong Yao.

**Writing – review & editing:** Xiaoyi Yang, Yinping Hu.

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
