## [Decision Letter · Decision Letter 0]

6 Jun 2025

Dear Dr. Yang,

Thank you for submitting your manuscript to PLOS ONE. After careful consideration, we feel that it has merit but does not fully meet PLOS ONE’s publication criteria as it currently stands. Therefore, we invite you to submit a revised version of the manuscript that addresses the points raised during the review process.

Please submit your revised manuscript by Jul 21 2025 11:59PM. If you will need significantly more time to complete your revisions, please reply to this message or contact the journal office at plosone@plos.org . A rebuttal letter that responds to each point raised by the academic editor and reviewer(s). You should upload this letter as a separate file labeled 'Response to Reviewers'.A marked-up copy of your manuscript that highlights changes made to the original version. You should upload this as a separate file labeled 'Revised Manuscript with Track Changes'.An unmarked version of your revised paper without tracked changes. You should upload this as a separate file labeled 'Manuscript'.

We look forward to receiving your revised manuscript.

Kind regards,

Frederick Quinn

Academic Editor

PLOS ONE

Journal Requirements:

2. Please include captions for your Supporting Information files at the end of your manuscript, and update any in-text citations to match accordingly. Please see our Supporting Information guidelines for more information: http://journals.plos.org/plosone/s/supporting-information .

3. Please remove all personal information, ensure that the data shared are in accordance with participant consent, and re-upload a fully anonymized data set.

Additional guidance on preparing raw data for publication can be found in our Data Policy (https://journals.plos.org/plosone/s/data-availability#loc-human-research-participant-data-and-other-sensitive-data) and in the following article: http://www.bmj.com/content/340/bmj.c181.long .

4. We note that there is identifying data in the Supporting Information file <Supplement material-data.xlsx>. Due to the inclusion of these potentially identifying data, we have removed this file from your file inventory. Prior to sharing human research participant data, authors should consult with an ethics committee to ensure data are shared in accordance with participant consent and all applicable local laws.

-Location data

Reviewers' comments:

Reviewer's Responses to Questions

**Comments to the Author**

1. Is the manuscript technically sound, and do the data support the conclusions?

Reviewer #1: Partly

Reviewer #2: Yes

2. Has the statistical analysis been performed appropriately and rigorously?

Reviewer #1: No

Reviewer #2: Yes

3. Have the authors made all data underlying the findings in their manuscript fully available?

Reviewer #1: Yes

Reviewer #2: Yes

4. Is the manuscript presented in an intelligible fashion and written in standard English?

Reviewer #1: No

Reviewer #2: Yes

Reviewer #1: I hope my suggestions will help with the manuscript improvement.

This manuscript should be thoroughly edited in English grammar to ensure clarity and readability for readers. Some important data is missing from the manuscript. The presentations of the results are confusing. Many issues need explanation as follows.

1. Abstract:

- Please state the evaluation methods of physical function in the abstract since it is one of the important parameters of this study.

- Please rewrite the conclusion in the abstract and the context. The cross-sectional study design cannot prove the causal relationship between parameters so the “indicators” or “predictors” cannot be applied in this study.

2. It is recommended that terminology be used consistently throughout the manuscript. For example, “physical balance function” or “Berg balance scale” or “body balance function”, those words are scattering thorough the context even though authors refer to the same things. The parameters “walking step” and 5 sit up times” are also presented as different terminology in other statements in the manuscript. This is very confusing. Please revise.

3. Methods

- All presentations are not consistent. Please state the operational criteria for categorization of MNA score, Berg Balance Scale, TUG, FTSST which implied in this study. In the results, authors mentioned “well-nourished” and “abnormal-nourished” but in the methods authors categorized as “well-nourished”, “risked malnutrition” and “malnourished”. Please consider revising the writing to ensure consistency with the way other variables are presented in the manuscript.

- Please indicate the citation for the cut-off points of each score

- Please state the statistical test for normalization testing

4. Results

- Please describe any concurrent infections (HIV, opportunistic infections, etc.) and details of comorbidities since all those factors involved in nutritional status and clinical outcomes

- A set of data must be presented either narrative or table but NOT both. Please rewrite the results and tables to NOT duplicate the same things.

- Please indicate the statistical tests applied for the P-value as a footnote (under each table)

- Please carefully check and specify the units of the parameters mentioned in the manuscript. There are missing units for the numbers in the results and tables.

- Again, all terminology must be used consistently throughout the manuscript

5. The discussion section should be improved. In my opinion, the authors tend to jump between variables without a clear structure and do not sufficiently refer to the actual findings of this study. A more focused and cohesive interpretation of the results is recommended. For example, abnormal BMI found in this study was 46.7% in the well-nourished groups and 45.7% in the abnormal-nourished groups, but authors stated “…the elderly in pulmonary TB with abnormal BMIs have worse nutritional status”. Another example, “YES” drinking status in the well-nourished groups was 26.65% and in in the abnormal-nourished groups was 13.7%, but author discussed “…alcohol consumption impairs the immune system, especially immune cells, limits the ability of monocytes to

regulate inflammation, increases inflammation in the patient's body, and increases energy expenditure, leading to a significant increase in the risk of malnutrition”. Please confirm.

Reviewer #2: Study about tuberculosis and it relation with physical function was interesting, although many variable should be considered for example nutritional intake with gold standar tool, and history of physical activity as well as musculoskeletal problem as a comorbidity.

**Do you want your identity to be public for this peer review?** For information about this choice, including consent withdrawal, please see our Privacy Policy

Reviewer #1: No

Reviewer #2: No

---

## [Author Response · Author response to Decision Letter 1]

21 Jul 2025

Point-by-point response

Journal Requirements:

1 Please ensure that your manuscript meets PLOS ONE's style requirements, including those for file naming.

Response: Thank you for your comment. We have thoroughly reviewed the manuscript and ensured that it fully complies with PLOS ONE's formatting and style requirements, including the appropriate file naming conventions.

2 Please include captions for your Supporting Information files at the end of your manuscript, and update any in-text citations to match accordingly.

Response: Thank you for your suggestion. We have added a dedicated section titled “Supporting Information” at the end of the manuscript, where each supporting file has been assigned an appropriate title (e.g., S1 Text: Patient Informed Consent Form and Research Questionnaires). In addition, we have updated all in-text citations to ensure consistency with the titles and numbering in the Supporting Information section.

3 Please remove all personal information, ensure that the data shared are in accordance with participant consent, and re-upload a fully anonymized data set.

Response: Thank you for your guidance. We have thoroughly reviewed our dataset and removed all personal information pertaining to the participants to ensure full anonymization. The shared data are now in complete accordance with the participant consent forms. A fully anonymized data set has been re-uploaded as requested. We confirm that no personally identifiable information remains within the dataset, and all necessary steps have been taken to protect participant confidentiality while maintaining the integrity of the research findings.

4 We note that there is identifying data in the Supporting Information file <Supplement material-data.xlsx>. Due to the inclusion of these potentially identifying data, we have removed this file from your file inventory. Prior to sharing human research participant data, authors should consult with an ethics committee to ensure data are shared in accordance with participant consent and all applicable local laws.

Response: Thank you for your comment. We sincerely apologize for the oversight. We have now fully anonymized the dataset by removing all personally identifiable information, in accordance with participant consent and ethical guidelines. A revised and fully anonymized version of the dataset has been re-uploaded. Additionally, prior to data sharing, we have consulted with our institutional ethics committee to ensure compliance with all applicable regulations and participant consent agreements. We confirm that the updated dataset contains no identifying information and is suitable for public sharing.

Reviewer 1:

1 Abstract:

- Please state the evaluation methods of physical function in the abstract since it is one of the important parameters of this study.

Response: Thanks for your constructive suggestion. We have stated the the evaluation methods of physical function in the abstract. This study assessed physical function using the Berg Balance Scale (BBS), the Timed Up and Go test (TUG), and the Five-Times-Sit-to-Stand Test (FTSST) on page 1.

- Please rewrite the conclusion in the abstract and the context. The cross-sectional study design cannot prove the causal relationship between parameters so the “indicators” or “predictors” cannot be applied in this study.

Response: Thank you for your question. We agree that the cross-sectional study design cannot prove the causal relationship between parameters. In light of this, we have revised the conclusion in the abstract and the context to reflect only the associations observed in our study.

Below is the revised version of the conclusion in the abstract:

Conclusions: Malnutrition is common in the elderly with pulmonary TB. Nutritional status in these patients is associated with the BBS scores, the TUG, and the FTSST (Page 1).

Below is the revised version of the conclusion in the context:

Our study shows that malnutrition is common in the elderly with pulmonary TB and and highlights a close association between nutritional status and physical function. Compared with well-nourished patients, the abnormal patients showed a significant decrease in the BBS scores, and a significant increase in the TUG and the FTSST. Healthcare professionals should focus on the nutritional status and physical function of patients, identify potential nutritional risk groups through early screening and assessment, and carry out early individualised nutritional interventions to improve patients' physical function and disease prognosis, and improve the quality of life of the elderly with pulmonary TB (Page 15).

2 It is recommended that terminology be used consistently throughout the manuscript. For example, “physical balance function” or “Berg balance scale” or “body balance function”, those words are scattering thorough the context even though authors refer to the same things. The parameters “walking step” and 5 sit up times” are also presented as different terminology in other statements in the manuscript. This is very confusing. Please revise.

Response: Thank you for your comment. We agree that consistent terminology is essential for clarity and scientific rigor. Accordingly, we have carefully reviewed the entire manuscript and standardized the terminology. We now uniformly use “Berg balance scale” (BBS) instead of varying expressions such as “physical balance function” or “body balance function”. Additionally, We uniformly use the Timed Up and Go test (TUG) instead of “walking times” and use the Five-Times-Sit-to-Stand Test (FTSST) instead of “5 sit up times”. We have also revised the entire manuscript accordingly.

3 Methods

- All presentations are not consistent. Please state the operational criteria for categorization of MNA score, Berg Balance Scale, TUG, FTSST which implied in this study. In the results, authors mentioned “well-nourished” and “abnormal-nourished” but in the methods authors categorized as “well-nourished”, “risked malnutrition” and “malnourished”. Please consider revising the writing to ensure consistency with the way other variables are presented in the manuscript.

Response: Thank you for your comment. We have supplemented the text with operational definitions and categorisation criteria for each research tools. In addition, the full scale is included in the appendix. Below is a detailed description of each tool. At the same time, we explained in the methods why the categories of “well-nourished”, “risked malnutrition” and “malnourished” were split into “well-nourished” and “abnormal-nourished” 2 groups for comparison. Based on clinical experience and relevant literature, all TB patients with nutritional risk should be assessed for their nutritional status. Therefore, in our study, patients with MNA scores <24 points (i.e. risked malnutrition: 17-24 and malnourished: <17 points), were categorised as abnormal-nourished group. This is mentioned in the abstract section (Page 1) and the introduction section of the Mini nutritional assessment (Page 4). In addition, we have made additional additions and inserted relevant citations in the operational criteria (Page 4-5).

Mini nutritional assessment (MNA)

The assessment was an internationally recommended nutritional screening tool for elderly patients [19–21]. It consisted of anthropometric indicators (4 entries), overall assessment (6 entries), dietary assessment (6 entries) and subjective rating (2 entries). There were 18 entries with a total score of 30.The judgement criteria was well nourished: ≥24 points, risked malnutrition: 17 to <24 points, malnourished: <17 points. The scale demonstrates effective screening performance for hospitalized elderly patients with chronic diseases [22]. Based on clinical experience and relevant literature [23, 24], all TB patients with nutritional risk should be assessed for their nutritional status. Therefore, in our study, patients with MNA scores <24 points (i.e. risked malnutrition: 17-24 and malnourished: <17 points), were categorised as abnormal-nourished group [25].

Berg Balance Scale (BBS)

The scale consisted of 14 balance-related entries as a means of assessing functional body balance in older adults [26]. A 4-point Likert scale (0 indicating the lowest level of functions and 4 indicating the highest) was used, with a total score of 56. 0-20 points (poor balance ability and can only sit in a wheelchair ), 21-40 points (balance ability and can assist with walking), 41-56 points (good balance ability and can walk independently) [27]. The Cronbach's alpha of the scale was 0.864.

Timed Up and Go test (TUG)

The test is recommended by the American Geriatrics Society as an indicator for assessing the patient's balance and walking function, especially for the elderly[28]. Test method: During the test, the patients wore their usual shoes, sat on a chair with armrests and a backrest, leaned their body on the back of the chair, and placed their hands on the armrests (seat height about 45cm, armrest height about 20cm). Stick or place a visible marker on the floor 3 metres away from the chair. When the command "start" was heard, the patient stood up firmly, followed the usual walking gait, walked towards the marker, turned around, returned to the chair, sat down, and leaned against the back of the chair. The researcher recorded the time it took the patient to complete the entire walking process, with the shorter time indicating the patient's ability to walk and balance better. The criteria were as follows: �10s was well, suggesting free movement; ≥10s was adverse, suggesting impaired movement [29].

Five-Times-Sit-to-Stand Test (FTSST)

This test, which recorded the time it took the subject to repeatedly stand up and sit down five times from a chair, was commonly used to assess lower limb strength and balance in the elderly [30]. Test method: The patients crossed their arms in front of their chests, looked straight ahead, and stood up and sat down 5 times as fast as they could from a 46cm high chair. All patients repeated the test 3 times with a 1 min break in between, and the average of the 3 times was taken as the final test result. Finally, the average test time for all patients was 10s. Thus, Time ≥10s was considered risk of falling; <10s was considered good balance and no risk [31].

- Please indicate the citation for the cut-off points of each score

Response: Thank you for your comment. We have added detailed citations for the cut-off points of each scoreto each research tools (MNA score, Berg Balance Scale, TUG, FTSST). (Page 4-5)

- Please state the statistical test for normalization testing

Response: Thank you for your suggestions. We have described the statistical methods for normality testing in the statistical analysis. All continuous variables were tested for normality using the Shapiro-Wilk test (α=0.05). Variables that conformed to normal distribution (Berg Balance Scale) were described using mean ± standard deviation, and t-tests were used for comparisons between groups. Variables that did not fit the normal distribution (Timed Up and Go test and Five-Times-Sit-to-Stand Test) were described by median [IQR], and comparisons between groups were made using the two related samples rank-sum test. Count data were described by n (%), and comparisons between groups were made using the χ² test. Correlations were analyzed using spearman's correlation analysis, and a difference of P<0.05 was considered statistically significant. (Page 6)

4 Results

- Please describe any concurrent infections (HIV, opportunistic infections, etc.) and details of comorbidities since all those factors involved in nutritional status and clinical outcomes

Response: Thank you for your comment. This is indeed a very important and worthwhile question. In this study, we did not systematically collect specific information about co-infections and comorbidities due to the limitations of the scope of data collection. This is a limitation of this study. In the revised manuscript, we will add a description of this limitation in the “Study limitations” section, and note that future studies should include such information more comprehensively to further clarify its impact on nutritional and clinical outcomes. Below is a detailed explanation of this issue.

Additionally, due to limitations in our data collection scope, we did not systematically gather specific information on co-infections and complications. Future studies could incorporate such data more comprehensively to better assess their impact on nutritional and clinical outcomes. (Page 14)

- A set of data must be presented either narrative or table but NOT both. Please rewrite the results and tables to NOT duplicate the same things.

Response: Thanks to your comments. We have adjusted the article to ensure that the same set of data is presented only in textual descriptions or tables to avoid repetitive expressions. Tables are used to present specific data, while the text only summarizes the main findings and does not repeat specific values to keep the presentation concise and logical.

- Please indicate the statistical tests applied for the P-value as a footnote (under each table)

Response: We have indicated the statistical tests performed on the p-values as footnotes under each table.

Table 1: There are no statistical tests in Table 1, only descriptive.

Table 2: P values represented whether the comparison between the two groups is significant. Count variables were tested by chi-square test. Continuous variables were tested with t-test and the two related samples rank-sum test. *P<0.05 was considered statistically significant.

Table 3: P values represented whether the comparison between the two groups is significant. P values were derived from the t-test (t) for normally distributed variables and the two related samples rank-sum test (Z) for non-normally distributed variables. *P<0.05 was considered statistically significant.

Table 4: P values represented Spearman's test. *P<0.05 was considered statistically significant.

- Please carefully check and specify the units of the parameters mentioned in the manuscript. There are missing units for the numbers in the results and tables.

Response: Thanks to your reminder, we have double-checked all the parameter units mentioned in the manuscript and added the missing units in the previous results and tables. It was ensured that all figures were accompanied by the correct units to ensure the accuracy and readability of the data.

- Again, all terminology must be used consistently throughout the manuscript

Response: We appreciate your comment. We have thoroughly checked and revised the manuscript to ensure all terminology is used consistently throughout.

5 The discussion section should be improved. In my opinion, the authors tend to jump between variables without a clear structure and do not sufficiently refer to the actual findings of this study. A more focused and cohesive interpretation of the results is recommended. For example, abnormal BMI found in this study was 46.7% in the well-nourished groups and 45.7% in the abnormal-nourished groups, but authors stated “…the elderly in pulmonary TB with abnormal BMIs have worse nutritional status”. Another example, “YES” drinking status in the well-nourished groups was 26.65% and in in the abnormal-nourished groups was 13.7%, but author discussed “…alcohol consumption impairs the immune system, especially immune cells, limits the ability of monocytes to

regulate inflammation, increases inflammation in the patient's body, and increases energy expenditure, leading to a significant increase in the risk of malnutrition”. Please confirm.

Response: Thank you very much for your valuable comments on our manuscript. We fully agree with your suggestion that the Discussion section needs to be further improved and structurally optimized. In response to the problems you have pointed out, we have already carried out a comprehensive sorting and rewriting of the Discussion section, aiming at clearer logic, more focused, and better analyzing and explaining with the actual results of the study (Page 10-13). The revised discussion section is as follows.

Potential factors influencing the nutritional status of elderly tuberculosis patients.

We compared the clinical

---

## [Decision Letter · Decision Letter 1]

8 Aug 2025

Thank you for submitting your manuscript to PLOS ONE. After careful consideration, we feel that it has merit but does not fully meet PLOS ONE’s publication criteria as it currently stands. Therefore, we invite you to submit a revised version of the manuscript that addresses the points raised during the review process.

Please submit your revised manuscript by Sep 22 2025 11:59PM. If you will need significantly more time than this to complete your revisions, please reply to this message or contact the journal office at plosone@plos.org . A rebuttal letter that responds to each point raised by the academic editor and reviewer(s). You should upload this letter as a separate file labeled 'Response to Reviewers'.A marked-up copy of your manuscript that highlights changes made to the original version. You should upload this as a separate file labeled 'Revised Manuscript with Track Changes'.An unmarked version of your revised paper without tracked changes. You should upload this as a separate file labeled 'Manuscript'.

We look forward to receiving your revised manuscript.

Kind regards,

Frederick Quinn

Academic Editor

PLOS ONE

Journal Requirements:

Reviewers' comments:

Reviewer's Responses to Questions

**Comments to the Author**

Reviewer #1: All comments have been addressed

Reviewer #2: All comments have been addressed

2. Is the manuscript technically sound, and do the data support the conclusions?

Reviewer #1: Partly

Reviewer #2: Yes

3. Has the statistical analysis been performed appropriately and rigorously?

Reviewer #1: Yes

Reviewer #2: Yes

4. Have the authors made all data underlying the findings in their manuscript fully available?

Reviewer #1: Yes

Reviewer #2: Yes

5. Is the manuscript presented in an intelligible fashion and written in standard English?

Reviewer #1: Yes

Reviewer #2: Yes

Reviewer #1: Almost all issues were clarified; however, some statements are still redundant with the tables presented in the results. The conclusion still overclaimed what was found in this study.

Reviewer #2: i accepted all revision that have been made by authors, despite all limitation this article could be improve reader insight about tuberculosis and it relationship

**Do you want your identity to be public for this peer review?** For information about this choice, including consent withdrawal, please see our Privacy Policy

Reviewer #1: No

Reviewer #2: **Yes: ** MOH SYAROFIL ANAM

---

## [Author Response · Author response to Decision Letter 2]

11 Aug 2025

Reviewer 1:

- Almost all issues were clarified; however, some statements are still redundant with the tables presented in the results. The conclusion still overclaimed what was found in this study.

Response: Thanks for your constructive suggestion. We fully agree with the reviewers' comments. To avoid repetition with the table content, we have reviewed the results section sentence by sentence, deleted redundant data repetition, and retained only a summary description of key trends and major findings. The revised text is more concise and highlights the key points. The relevant adjustments are clearly marked in “revision mode” in the revised draft. Additionally, We agree that the original wording may have overstated the study’s findings. Accordingly, we have revised the conclusion to present the results more cautiously, clarifying that our findings suggest an association rather than causation and emphasizing the need for further research. The modified conclusion now reads

“Our study shows that malnutrition is prevalent among in the elderly with pulmonary TB and may be associated with poorer physical function. Compared with well-nourished patients, the abnormal patients tended to have lower BBS scores and higher TUG and FTSST, though further research is needed to confirm causality. These findings highlight the potential importance of nutritional screening and assessment in this population. Early identification of malnutrition and tailored interventions may help improve physical function and quality of life, but additional studies are warranted.” (Page 15)

We hope these revisions address the reviewer’s concern and better reflect the scope of our study’s implications. Thank you for your constructive critique, which has strengthened our manuscript.

Reviewer 2:

- I accepted all revision that have been made by authors, despite all limitation this article could be improve reader insight about tuberculosis and it relationship

Response: We sincerely appreciate your time and constructive feedback, which has significantly improved the quality of our manuscript. We are grateful for your acknowledgment of the revisions and your encouraging comment regarding the article’s potential to enhance readers’ understanding of tuberculosis and its associated factors.

As you highlighted, while our study has limitations, we hope it provides valuable insights and serves as a foundation for future research in this field. Thank you once again for your thoughtful review and support. We look forward to contributing further to the scientific discourse on this important topic.

---

## [Decision Letter · Decision Letter 2]

20 Aug 2025

A correlation analysis between nutritional status and physical function in the elderly with pulmonary tuberculosis

PONE-D-25-13007R2

Dear Dr. Yang,

We’re pleased to inform you that your manuscript has been judged scientifically suitable for publication and will be formally accepted for publication once it meets all outstanding technical requirements.

Kind regards,

Frederick Quinn

Academic Editor

PLOS ONE

Additional Editor Comments (optional):

Reviewers' comments:

Reviewer's Responses to Questions

**Comments to the Author**

Reviewer #1: All comments have been addressed

Reviewer #2: All comments have been addressed

2. Is the manuscript technically sound, and do the data support the conclusions?

Reviewer #1: Yes

Reviewer #2: Yes

3. Has the statistical analysis been performed appropriately and rigorously?

Reviewer #1: Yes

Reviewer #2: Yes

4. Have the authors made all data underlying the findings in their manuscript fully available?

Reviewer #1: Yes

Reviewer #2: Yes

5. Is the manuscript presented in an intelligible fashion and written in standard English?

Reviewer #1: Yes

Reviewer #2: Yes

Reviewer #1: All issues were addressed. The conclusion were included what's found in the context. The current version is acceptable.

Reviewer #2: all point have been clearly revised

this article add value of information regarding tuberculosis especially in elderly

**Do you want your identity to be public for this peer review?** For information about this choice, including consent withdrawal, please see our Privacy Policy

Reviewer #1: No

Reviewer #2: **Yes: ** MOH SYAROFIL ANAM

---

## [Editor Report · Acceptance letter]

PONE-D-25-13007R2

PLOS ONE

Dear Dr. Yang,

I'm pleased to inform you that your manuscript has been deemed suitable for publication in PLOS ONE. Congratulations! Your manuscript is now being handed over to our production team.

Kind regards,

on behalf of

Dr. Frederick Quinn

Academic Editor

PLOS ONE